# Cessation of Face Mask Use after COVID-19 Vaccination in Patients with Diabetes: Prevalence and Determinants

**DOI:** 10.3390/ijerph20042768

**Published:** 2023-02-04

**Authors:** Hid Felizardo Cordero Franco, Ana María Salinas Martínez, Diana Laura Martínez Martínez, Blanca Reyna Santiago Jarquin, Francisco Javier Guzmán de la Garza

**Affiliations:** 1Epidemiologic and Health Services Research Unit/CIBIN, Mexican Institute of Social Security, Monterrey 64360, Mexico; 2School of Public Health and Nutrition, Autonomous University of Nuevo Leon, Monterrey 64460, Mexico; 3Vice-Rectory of Health Sciences, University of Monterrey, San Pedro Garza García 66238, Mexico; 4Family Medicine Clinic No. 26, Mexican Institute of Social Security, Monterrey 64360, Mexico; 5School of Medicine, Autonomous University of Nuevo Leon, Monterrey 64460, Mexico

**Keywords:** COVID-19, vaccines, masks, health belief model, health behavior

## Abstract

Studies on the cessation of face mask use after a COVID-19 vaccine in patients with diabetes are not available, despite their greater predisposition to complications. We estimated the prevalence of cessation of face mask use after receiving the COVID-19 vaccine in patients with diabetes and identified which factor was most strongly associated with non-use. This was a cross-sectional study in patients with diabetes 18–70 years with at least one dose of vaccine against COVID-19 (*n* = 288). Participants were asked to respond face-to-face to a questionnaire in a primary care center. Descriptive statistics, chi-square tests, and multivariate binary logistic regression were used for analyzing the association between vulnerability, benefits, barriers, self-efficacy, vaccine expectations (independent variables), and cessation of use (dependent variable), controlling for sociodemographic, smoking, medical, vaccine, and COVID-19 history. The prevalence of cessation of face masks was 25.3% (95% CI 20.2, 30.5). Not feeling vulnerable to hospitalization increased the odds of non-use (adjusted OR = 3.3, 95% CI 1.2, 8.6), while perceiving benefits did the opposite (adjusted OR = 0.4, 95% CI 0.2, 0.9). The prevalence was low, and only two factors were associated with the cessation of face mask use after COVID-19 vaccination in patients with type 2 diabetes.

## 1. Introduction

During the COVID-19 pandemic, the use of face masks showed to be effective in reducing the spread of the disease and, secondarily, in decreasing hospitalization and mortality due to the disease [1,2,3,4,5]. On the other hand, the greatest benefit of the vaccines was observed in the reduction in disease severity and, therefore, in the decreased need for hospital care and death from COVID-19 [6,7]. Mexico was one of the countries with the highest percentage of face mask use; 9 out of 19 Mexicans wore them as of May 2022 (one year after mass vaccination started) [8]. Moreover, 12,409,086,286 vaccines had been applied worldwide as of August 2022; 209,673,612 correspond to Mexico [9], and 63% of Mexican adults had completed a full vaccination scheme [10]. Patients with diabetes were at increased risk of severe COVID-19 disease and had priority to be vaccinated [11,12,13,14,15]. However, institutions such as the World Health Organization [16], the Centers for Disease Control and Prevention [17], and the American Diabetes Association [18] maintained the recommendation of keeping a healthy distance and the use of face masks, even after vaccination, because the risk of severe illness and death continued to exist.

Various theoretical models have been proposed to explain health behaviors. The Health Belief Model considers vulnerability, benefits, and barriers [19,20,21], which together with self-efficacy help explain the use of COVID-19 preventive measures [22,23,24,25,26]. Expectations also play a key role in maintaining a healthy behavior, and continuing the use of face masks may depend on the expected benefits of the vaccine; if the result does not occur in the way it was anticipated, the behavior may be abandoned or replaced [27]. Furthermore, unrealistic expectations can be a source of the false security of not getting sick, accompanied by wrong decisions in the use of preventive measures. Hence, expecting 100% protection against COVID-19 infection by the vaccine could lead to perceived invulnerability and the cessation of the use of face masks. The intention to maintain the behavior of mask use after vaccination has not been consistent in general populations. Studies from China [28,29], Italy [30], the United Kingdom [31], and the United States [32,33] show that vaccinated people tend to maintain preventive recommendations, suggesting that the perception of risk of the disease remains. However, some others indicate the opposite [34,35,36,37]. A survey from the US revealed that 5–6 out of 10 adults intended to continue wearing a face mask or using other preventive measures after receiving the first dose [38]. Studies on the cessation of the use of masks in patients with diabetes after being vaccinated against COVID-19 are not available, despite the importance of discontinuing the use of face masks in any population, and even more, in patients with diabetes because of their greater predisposition to complications and risk of dying from COVID-19 [39]. Although today the use of masks is no longer mandatory in Mexico and other regions, the study of the cessation of mask use still provides essential and valuable information. Lessons from the past allow for correct planning. The recognition of factors that influenced the initiation and maintenance of the use of masks makes it possible to foresee strategies that would favor their utilization in the event of a resurgence of this or another infectious disease that warrants this type of measure. The objective of the present study was to estimate the prevalence of cessation of face mask use after receiving the COVID-19 vaccine in patients with diabetes. Additionally, to identify which factor (vulnerability, benefits, barriers, self-efficacy, or vaccine’s expectations) was most strongly associated with non-use.

## 2. Materials and Methods

This was a cross-sectional study conducted from September 2021 to February 2022 in Monterrey, Mexico. The study population consisted of patients with diabetes between 18 and 70 years with at least one dose of vaccine against COVID-19 (n = 288). The sampling technique was non-random. The participants were invited to respond to a structured questionnaire in a primary care center of the Mexican Institute of Social Security. The face-to-face survey lasted 10 to 15 min and was conducted in a private room during morning and evening shifts (8 a.m. to 8 p.m.). Collaboration was entirely voluntary, and withdrawal was allowed at any time without the need to justify the decision; no participant withdrew. The sample size was enough for 5% precision and a 95% confidence level because the post-vaccine face mask cessation prevalence was 25%, using the free statistical software Epidat 3.1 [40]; this was corroborated with the formula [41]:$$\frac{N={\left(Z\alpha \right)}^{2}\left(p\right)\left(q\right)}{\delta^{2}}$$
where, *N =* sample size, *Zα =* 1.96, *p =* 0.25, *q =* 0.75, *δ =* 0.05.

The research protocol was approved by the Local Committee of Ethics and Health Research (R-2021-1909-104). Written informed consent was obtained from all participants, and the Declaration of Helsinki for Research on Human Subjects’ Guidelines were followed [42].

### 2.1. Dependent Variable

#### Cessation of Face Mask Use after Receiving the COVID-19 Vaccine

Five questions were included. They were adapted from Schoeni et al. [43]. We identified the frequency of face mask use before receiving the vaccine: (1) inside the car, the bus, or the subway, (2) while walking outdoors, (3) while talking inside the house with someone with whom he/she does not live, (4) while talking outside the house with someone with whom he/she does not live, and (5) while waiting for food in a restaurant (1 = never, 5 = always); 5 items, Cronbach’s alpha = 0.80. The questions were asked again, but now after receiving the COVID-19 vaccine (5 items, Cronbach’s alpha = 0.85). Pre- and post-vaccine responses to each question were paired. If the pre-vaccine response was always or almost always and the post-vaccine response was sometimes, rarely, or never worn, the face mask use was coded as 1 = cessation of use. If the opposite occurred or remained the same, the face mask use was coded as 0 = no cessation of use. The positive items were then summed (possible range of 0 to 5). A score greater than or equal to 3 defined the category of overall cessation of face mask use.

### 2.2. Independent Variables

#### 2.2.1. COVID-19 Vulnerability before and after Receiving the COVID-19 Vaccine

Four questions were included. They were adapted from the literature [44,45,46,47,48]. We identified vulnerability before receiving the vaccine: (1) for getting infected, (2) for developing symptoms, (3) for being hospitalized, and (4) for dying from the disease (1 = not any, 5 = very high; 4 items, Cronbach’s alpha = 0.92). The questions were asked again, but now after receiving the COVID-19 vaccine (4 items with Cronbach’s alpha = 0.88). Pre- and post-vaccine responses to each question were paired. If the pre-vaccine response was high or very high and the post-vaccine response was low, very low, or not any, the vulnerability was coded as 1 = stopped feeling vulnerable. If the opposite occurred or remained the same, the vulnerability was coded as 0 = did not stop feeling vulnerable. The positive answers were not added up; they were analyzed individually.

#### 2.2.2. Use of Face Masks Benefits

Ten questions were included. They were adapted from the Health Belief Model [19,20,49]. We identified the benefits of wearing face masks (e.g., it keeps you safe from coronavirus) (−1 = no, 0 = do not know, +1 = yes; 10 items, Cronbach’s alpha = 0.78). The positive responses were coded as 1 = did perceive the benefit; negative and do not know responses were regrouped and coded as 0 = did not perceive the benefit. The positive items were then summed (possible range of 0 to 10). A score greater than or equal to 6 defined the category of the overall perception of the benefits of wearing face masks.

#### 2.2.3. Use of Face Masks Barriers

Ten questions were included. They were adapted from the Health Belief Model [19,20,49] for identifying barriers to wearing face masks (e.g., the use of a face mask is uncomfortable) (−1 = no, 0 = do not know, +1 = yes; 10 items, Cronbach’s alpha = 0.72). The positive responses were coded as 1 = did perceive the barrier; negative and do not know responses were regrouped and coded as 0 = did not perceive the barrier. The positive items were then summed (possible range of 0 to 10). A score greater than or equal to 6 defined the category of the overall perception of barriers to wearing face masks.

#### 2.2.4. Use of Face Masks Self-Efficacy

Seven questions were included for identifying the self-confidence for wearing face masks: (1) While partying or gathering with family/friends, (2) while waiting for food in a restaurant, (3) while being inside of a crowded place, (4) despite keeping it on it is annoying, (5) despite being difficult to breathe with the mask on, (6) despite keeping it on it is uncomfortable, and (7) despite there being people who think face masks are useless (0 = not any, 4 = very high; 7 items, Cronbach’s alpha = 0.92). The answers high and very high were regrouped and coded as 1 = did perceive self-efficacy. The answers low, very low, and not any were regrouped and coded as 0 = did not perceive self-efficacy. The positive items were then summed (possible range of 0 to 7). A score greater than or equal to 5 defined the category of the overall perception of self-efficacy for wearing face masks.

#### 2.2.5. Vaccine’s Realistic Expectations

Three questions were included for identifying the realistic expectations of the COVID-19 vaccine: (1) It will reduce virus transmission, (2) it will reduce the risk of hospitalization, and (3) it will reduce the risk of dying from the disease (−1 = no, 0 = do not know, +1 = yes; 3 items, Cronbach’s alpha = 0.52). The positive responses were coded as 1 = did have realistic expectations; negative and do not know responses were regrouped and coded as 0 = did not have realistic expectations. The positive items were then summed (possible range of 0 to 3). A score greater than or equal to 2 defined the category of the overall perception of vaccines’ realistic expectations.

#### 2.2.6. Vaccine’s Unrealistic Expectations

Three questions were included for identifying unrealistic expectations of the COVID-19 vaccine: (1) It will totally prevent contagion, (2) it will make it possible to stop using the mask, and (3) it will make it possible to stop using antibacterial gel (−1 = no, 0 = do not know, +1 = yes; 3 items, Cronbach’s alpha = 0.45). The positive responses were coded as 1 = did have unrealistic expectations; negative and do not know responses were regrouped and coded as 0 = did not have unrealistic expectations. The positive items were then summed (possible range of 0 to 3). A score greater than or equal to 2 defined the category of the overall perception of vaccines’ unrealistic expectations.

#### 2.2.7. Other Variables That Can Affect Face Mask Use

Self-report of medical history (hypertension, cardiovascular disease, other), self-report of personal or family history of COVID-19, number of doses and vaccine type (virus vector, mRNA, inactivated virus), smoking, age, sex, schooling, marital status, and occupation.

### 2.3. Procedures

Data were collected by trained personnel (a medical resident and a medical intern). The questionnaire was designed in Spanish. All items were subject to content validity, and special attention was paid to avoiding ambiguity and technical vocabulary. The reliability results have already been provided. Pre-test and pilot tests were carried out to verify the clarity and ease of understanding. The final version is available in Appendix A.

### 2.4. Statistical Analysis

Measures of central tendency and dispersion were estimated for the numerical variables, and proportions and 95% confidence intervals (CI) for the categorical variables. The chi-square test was used for analyzing the association between study factors (categorized as yes or no) and cessation of face mask use (categorized as yes or no). Multivariate binary logistic regression was used for estimating odds ratios (OR) and 95% CI. Vulnerability, benefits, barriers, self-efficacy, and vaccine expectations represented the independent variables; cessation of use was the dependent variable; and sociodemographic, smoking, medical, vaccine, and COVID-19 history were the control variables. Independent and control variables were entered in a single step (enter method). A *p*-value < 0.05 was considered significant.

## 3. Results

### 3.1. Descriptive Statistics

The mean age was 57.9 ± 9.5 years. More than half of the participants were female with secondary and higher education. The most common medical history was hypertension, and a high percentage had had COVID-19 him/herself or his/her family. Nearly 94% of patients had received two or more doses of the COVID-19 vaccine at the time of the survey (Table 1). There was no significant difference in the type of vaccine and number of doses between men and women (*p* = 0.20 and 0.21, respectively).

The overall prevalence of cessation of face mask use was 25.3% (95% CI 20.2, 30.5), and most people stopped wearing a mask in their homes (Figure 1).

High and very high vulnerability rates were 80% or higher before receiving the vaccine and they ranged from 27% to 51%, after vaccination (Figure 2).

The overall frequency of face mask benefits and barriers was 89.2% and 25.3%, respectively (95% CI 85.7, 92.8, and 95% CI 20.3, 30.4, respectively), whereas the overall frequency of high and very high self-efficacy was 96.2% (95% CI 94.0, 98.4). The overall vaccine’s realistic expectations frequency was 80.9% and that of unrealistic expectations was 14.6% (95% CI 76.4, 85.4, and 10.5, 18.7, respectively) (Figure 3, Figure 4, Figure 5 and Figure 6).

Additionally, the main variables were analyzed according to the number of vaccine doses received (one dose versus two or more doses), without finding significant differences (Appendix A).

### 3.2. Factors Associated with Cessation of Face Mask Use

Not feeling vulnerable after vaccination for getting the infection (unadjusted OR = 2.6, 95% CI 1.5, 4.5, *p* < 0.001), developing symptoms (unadjusted OR = 2.1, 95% CI 1.3, 3.7, *p* < 0.001), being hospitalized (unadjusted OR = 4.4, 95% CI 2.3, 8.3, *p* < 0.0001), and dying from COVID-19 (unadjusted OR = 3.1, 95% CI 1.8, 5.5, *p* < 0.0001) were associated with cessation of face mask use at the univariate level. Only two factors were related at the multivariate level, discontinuation of feeling vulnerable for hospitalization increased the odds of non-use after vaccination (adjusted OR = 3.3, 95% CI 1.2, 8.6), and the perception of face mask benefits reduced them (adjusted OR = 0.4, 95% CI 0.2, 0.9) (Table 2).

## 4. Discussion

This study focused on the cessation of face mask prevalence and factors associated with wearing a face mask after receiving the COVID-19 vaccine in patients with type 2 diabetes. We found that one in four patients discontinued the use of face masks after vaccination, and the prevalence varied from 20 to 50%. An Israeli study found 10% in the general population [34], a Chinese study identified 22% in a student population [35], and an Ethiopian study documented 31% in a population of health workers [36]. The analysis by location showed cessation was highest at home and lowest in public places. Maybe because at the time of the study, the use of face masks was mandatory in most public places. Burger et al. [50] indicated the use of face masks had become a new social norm that helped protect others. As of today, many countries have suspended the mandatory use of face masks. Recognizing what favors the use and maintenance of the face mask is still valuable since the COVID-19 pandemic is not over yet. Moreover, it is useful for health authorities who wish to promote its use to prevent the spread of other highly transmissible respiratory diseases such as influenza. A notable increase in cases of influenza is expected due to the relaxation of the use of face masks [51]. We must not forget that the severity of influenza may be greater in people with diabetes [52]. Regarding vulnerability, at least 8 out of 10 participants felt very susceptible to the disease before getting vaccinated, similar to other reports [43,45,46,47,48,53]. However, more than 50% stopped feeling vulnerable to dying or being hospitalized after the vaccine. Furthermore, discontinuation of feeling vulnerable for hospitalization increased three times the possibility of cessation of face mask use. It is understandable to observe relaxation in those who feel protected by the vaccine [54]. The role played by post-vaccine susceptibility in preventive behaviors has been controversial. Some reports have confirmed this relationship [25,36,55,56,57,58], but others have not [25,36,55,56,57,58]. More research is needed to analyze the impact of this component of the health belief model on the maintenance of health measures during an epidemic.

Due to their effectiveness, the use of masks during the COVID-19 pandemic was strongly recommended for the protection of oneself and others [59]. Fortunately, the global prevalence of perceived benefits was high: 9 out of 10 participants perceived advantages in the use of face masks. The perception of individual benefits varied from 74 to 97%, figures that were considerably higher than those reported by Keller et al. [60] of 46%. We also identified that the perception of benefits reduced the possibility of cessation of face mask use by 64%, which was in accordance with what was documented in a Chinese study [56]. Only one in four patients perceived barriers to wearing a face mask. The most frequently mentioned barrier had to do with the hot weather, typical of the summer in the city where the study was carried out. Discomfort has been described as a factor against wearing a face mask [56]. The cost and not recognizing the need for its use when there are no people around are other known barriers [50,61], which were also present but less frequently. Notably, the perception of barriers was not associated with the cessation of use or, contrary to what was expected [26,50,56,58], self-efficacy either. One possible explanation is that the barriers were overcome by the benefits [62] and that the prevalence of self-efficacy was high: 9 out of 10 participants perceived themselves as very confident in wearing a face mask. Finally, we anticipated a greater interruption of use due to high expectations of the effectiveness of the vaccine to reduce the risk and spread of the disease [63], but this did not happen. A high percentage of participants correctly expected such outcomes. Mask fatigue or a lack of energy to wear a face mask after a long time has been described [64], and although 7 out of 10 people expected that the vaccine would completely prevent them from getting the infection, a reduced frequency of respondents expected to stop wearing the mask thanks to the vaccine. Thus, the population with diabetes was correctly aware that the scope of the vaccine was limited and that there was a need to continue with self-care measures [16,17,18]. Health authorities must recognize which factors influence the maintenance of the use of preventive measures to focus on health education strategies that are useful not only to deal with emerging diseases but also with chronic-degenerative diseases [65,66].

Limitations. All the participants were residents of the Monterrey urban area; therefore, it is not possible to generalize the results to individuals who live in non-urban areas due to possible socioeconomic differences. Furthermore, they were primary care users. The results cannot be applied to those who are receiving secondary and tertiary care, because advanced complications might influence perceptions and expectations. In the future, it would be desirable to include patients with these characteristics. Another limitation corresponds to the cross-sectional design of the study; a longitudinal approach will be required in the future for definitive association results. Finally, the study focused only on patients with diabetes due to their higher risk and poor COVID-19 prognosis. It is desirable to replicate the methodology in patients with other health conditions and to analyze determinants of the use of other preventive measures in populations at higher risk.

## 5. Conclusions

The prevalence of face mask cessation was low after COVID-19 vaccination in patients with type 2 diabetes. Two factors were associated: not feeling vulnerable to being hospitalized increased the odds of less use, and the perception of the benefits diminished them. The study of factors that favor the maintenance of preventive measures is important for all types of diseases, infectious and non-infectious. Undoubtedly, the recognition of what determines the use of face masks would allow specific promotion campaigns to be carried out with a greater possibility of success in preventing the spread of communicable diseases.

## Figures and Tables

**Figure 1 ijerph-20-02768-f001:**
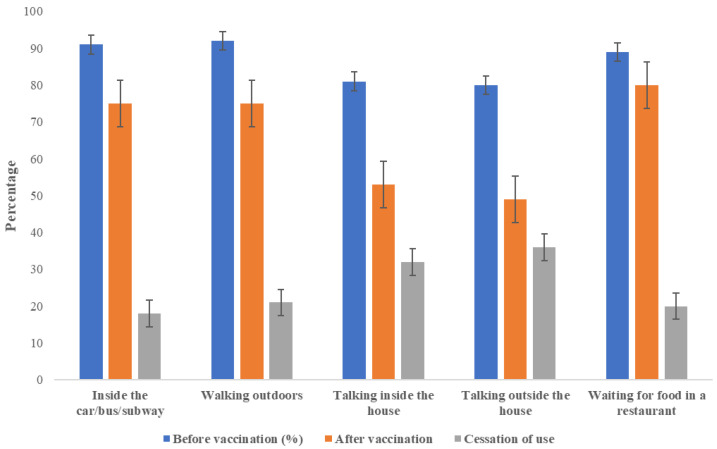
Mask use (always or almost always) before and after receiving the COVID-19 vaccine and cessation of use after vaccination (change from always or almost always to sometimes, rarely, or never) in patients with diabetes (n = 288).

**Figure 2 ijerph-20-02768-f002:**
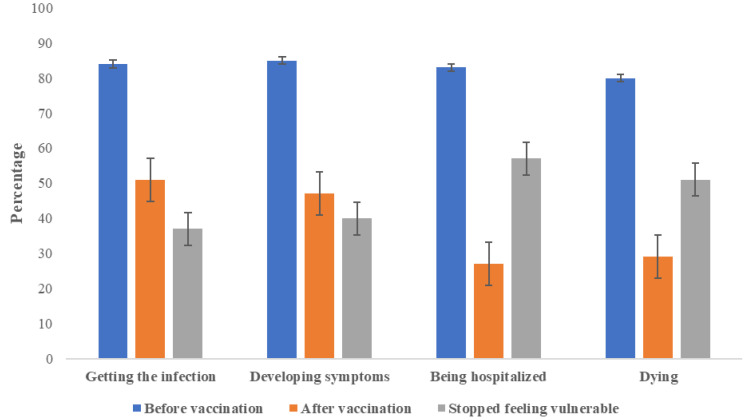
Vulnerability (high and very high) before and after receiving the COVID-19 vaccine and not feeling vulnerable after vaccination (change from very high or high to low, very low or not any) in patients with diabetes (n = 288).

**Figure 3 ijerph-20-02768-f003:**
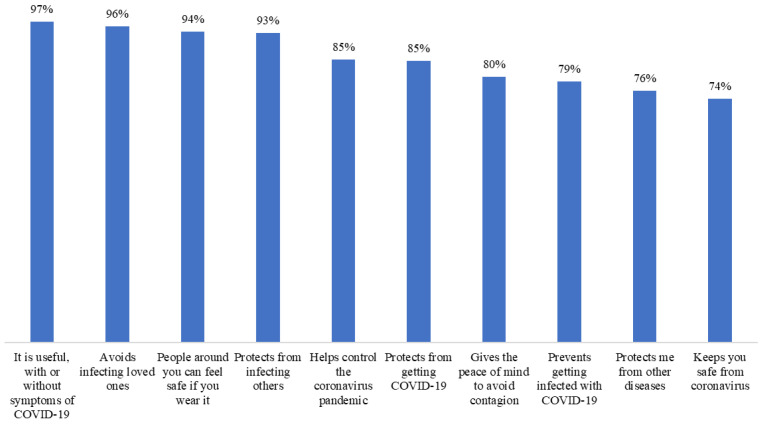
Perception of benefits of wearing face masks in patients with diabetes (n = 288).

**Figure 4 ijerph-20-02768-f004:**
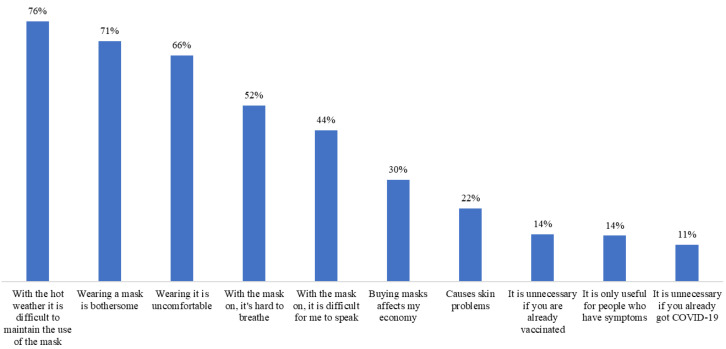
Perception of barriers to wearing face masks in patients with diabetes (n = 288).

**Figure 5 ijerph-20-02768-f005:**
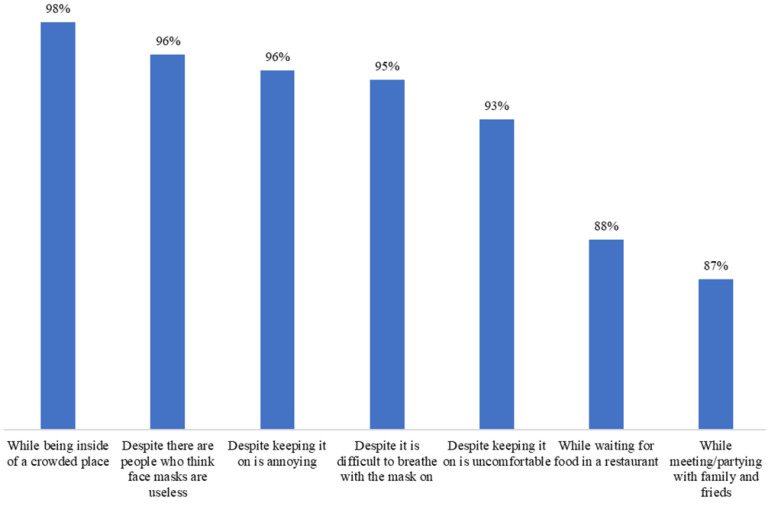
Self-efficacy (high or very high) of wearing face masks in patients with diabetes (n = 288).

**Figure 6 ijerph-20-02768-f006:**
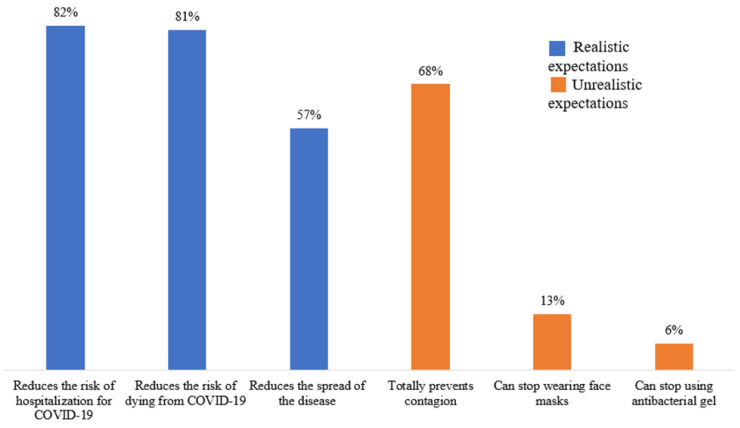
Vaccine’s realistic and unrealistic expectations in patients with diabetes (n = 288).

**Table 1 ijerph-20-02768-t001:** Sociodemographic, smoking, medical, and COVID-19 history in patients with type 2 diabetes with at least one dose of vaccine against COVID-19 (n = 288).

Characteristic	Frequency n (%)
Sociodemographic	
Sex, female	187 (64.9)
Marital status, lives with a partner	227 (78.8)
Occupation, economically active	112 (38.9)
Schooling, high school, or less	151 (52.4)
Smoking	125 (44.4)
Medical history	
Hypertension	208 (72.2)
Cardiovascular disease	35 (12.2)
Chronic obstructive pulmonary disease	25 (8.7)
Immunosuppression	7 (2.4)
Chronic kidney disease	7 (2.4)
Cancer	5 (1.7)
Type of vaccine	
AZD-1222 (virus vector)	184 (63.9)
BNT162b2 (mRNA)	53 (18.4)
CoronaVac (inactivated virus)	17 (5.9)
mRNA-1273 (mRNA)	17 (5.9)
Ad5-nCoV (virus vector)	4 (1.4)
JNJ-78436735 (virus vector)	2 (0.7)
Unknown	11 (3.8)
Number of doses	
One	18 (6.3)
Two	153 (53.1)
Three or more	117 (40.6)
COVID-19 infection	
Before the 1st dose	82 (70.1)
Before the 2nd dose	1 (0.9)
After the 2nd dose	34 (29.1)
Needed hospitalization	6 (5.1)
A family member got sick with COVID-19	243 (84.4)
A family member died because of COVID-19	1 (0.4)

**Table 2 ijerph-20-02768-t002:** Multivariate analysis of the vulnerability, benefits, barriers, self-efficacy, vaccine expectations, and cessation of use of face masks.

	Cessation of Use		Unadjusted OR	Adjusted OR ^a^
	Yes (n = 73) n (%)	No (n = 215) n (%)	(95% CI)	(95% CI)
Not feeling vulnerable after vaccination				
For getting the infection	39 (53.4)	66 (30.7)	2.59 (1.51, 4.45)	2.07 (0.75, 5.70)
For developing symptoms	39 (53.4)	75 (34.9)	2.14 (1.25, 3.66)	0.93 (0.34, 2.54)
For being hospitalized	59 (80.8)	105 (48.8)	4.41 (2.34, 8.31)	3.26 (1.23, 8.60)
For dying from the disease	52 (71.2)	95 (44.2)	3.13 (1.77, 5.52)	1.18 (0.49, 2.82)
Perception of benefits	61 (83.6)	196 (91.2)	0.49 (0.23, 1.06)	0.36 (0.15, 0.86)
Perception of barriers	13 (17.8)	60 (27.9)	0.56 (0.29, 1.09)	0.62 (0.30, 1.29)
High and very high self-efficacy	71 (97.3)	206 (95.8)	1.55 (0.33, 7.35)	1.17 (0.20, 6.78)
Vaccine’s realistic expectations	59 (80.8)	174 (80.9)	0.99 (0.51, 1.94)	0.91 (0.42, 1.97)
Vaccine’s unrealistic expectations	10 (13.7)	32 (14.9)	0.91 (0.43, 1.93)	1.15 (0.49, 2.70)

^a^ Adjusted for sociodemographic, smoking, medical, number of vaccine doses, and COVID-19 history. OR: odds ratio. CI: confidence interval.

## Data Availability

Not applicable.

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
