# Peer review of "Cessation of Face Mask Use after COVID-19 Vaccination in Patients with Diabetes: Prevalence and Determinants"

_ijerph, 2023, doi:10.3390/ijerph20042768_

Round 1

Reviewer 1 Report (Previous Reviewer 1)

The authors provide a well-written research article concerning the cessation of face-mask-wearing in the population. They address the importance of wearing face masks, especially among more vulnerable populations like people with co-existing conditions including diabetes.

The authors mention that the participants were patients who had received at least one vaccine. It will help to further separate out data based on the number of vaccine shots received.

The research findings are interesting and health implications are beneficial.

Author Response

Comments from reviewer:

The authors provide a well-written research article concerning the cessation of face-mask-wearing in the population. They address the importance of wearing face masks, especially among more vulnerable populations like people with co-existing conditions including diabetes.

The authors mention that the participants were patients who had received at least one vaccine. It will help to further separate out data based on the number of vaccine shots received.

The research findings are interesting and health implications are beneficial.

Answer: thank you very much for your comments. In response to your suggestion, we divided the main variables according to the number of vaccine doses received (one dose versus two or more doses). No significant differences were found. This information has been added in the Results section and displayed in the Supplementary file S2 for your consideration. Also, as we mention in Table 2, the number of vaccine doses was one of the control variables.

This manuscript is a resubmission of an earlier submission. The following is a list of the peer review reports and author responses from that submission.

Round 1

Reviewer 1 Report

Line 89, the statement is unclear. 

95, 96 – The method of coding sounds confusing. Considering the fact that the end responses could be different, why is coding everything the same? Sometimes and rarely are not the same so should not be counted as cessation of wear. 

Was pairing done for any other responses? 

Attach the materials used for the questions/interviews in the document or as supplementary.

Line 160, ease of understanding 

Line 175. Is this with respect to males only or female participants as well? It states 'Himself or his family'.

Was covid vaccination percentage higher in males or females? Stratify the data. 

181: rephrase…most people stop wearing in their homes.

Was there data collected on factors that would influence the participants to continue wearing facemasks? In most places, the mask is no longer mandatory so some information in that regard would make the context even more relevant to the current situation.

Author Response

Comments and Suggestions for Authors (Reviewer 1):

  1. Line 89, the statement is unclear.

Response: that statement was rewritten. If it remains unclear, please let us know.

  1. 95, 96 – The method of coding sounds confusing. Considering the fact that the end responses could be different, why is coding everything the same? Sometimes and rarely are not the same so should not be counted as cessation of wear.

Response: The reviewer is correct, the expression has been corrected, now it says reduction/cessation of use.

  1. Was pairing done for any other responses?

Response: Only the face mask use and vulnerability responses were paired. Not the perception of benefits, barriers, self-efficacy, and vaccine’s expectations responses.

  1. Attach the materials used for the questions/interviews in the document or as supplementary.

Response: The questionnaire is now available as supplementary material.

  1. Line 160, ease of understanding.

Response: Done.

  1. Line 175. Is this with respect to males only or female participants as well? It states 'Himself or his family'.

Response: The sentence was corrected. It referred to both, male and female participants.

  1. Was covid vaccination percentage higher in males or females? Stratify the data.

Response: All participants had the vaccine, as it was an inclusion criterion (lines 79 and 80). Data could only be stratified by type of vaccine and number of doses. This information is now available (lines 179 and 180).

  1. 181: rephrase…most people stop wearing in their homes.

Response: Done.

  1. Was there data collected on factors that would influence the participants to continue wearing facemasks?

Response: Yes. We considered medical history, personal or family history of COVID-19, smoking, age, sex, and schooling, among others (line 154). In addition to perception of vulnerability, benefits, barriers, self-efficacy, and vaccine expectations).

  1. In most places, the mask is no longer mandatory so some information in that regard would make the context even more relevant to the current situation.

Response: We added information according to the current pandemic situation (lines 232-238).

Reviewer 2 Report

The authors have co-related the usage of face masks to exposure to covid symptoms within a specific geographical area in patients with diabetes. The data is obtained from the survey among these patients.

-       As the author itself states, the limitation of the study is that it is confined to a specific geographical area. The survey is highly influenced by the socio-economic status of the area.

-       Also the data is confined to patients with diabetes. Therefore it is difficult to generalize any of the findings.

Author Response

Comments and Suggestions for Authors (Reviewer 2):

The authors have co-related the usage of face masks to exposure to covid symptoms within a specific geographical area in patients with diabetes. The data is obtained from the survey among these patients.

  1. As the author itself states, the limitation of the study is that it is confined to a specific geographical area. The survey is highly influenced by the socio-economic status of the area.

Response: thanks for your comment. We added it as a limitation of the study.

  1. Also, the data is confined to patients with diabetes. Therefore, it is difficult to generalize any of the findings.

Response: Yes, we were interested in these patients due to their higher risk and poor prognosis. But the methodology can be used in people with other health conditions. Or, to analyze determinants of the use of other preventive measures in populations at higher risk.

Round 2

Reviewer 2 Report

The authors have revised their manuscript in addressing the concerns of the reviewers.